# Vitamin E-Loaded PLA- and PLGA-Based Core-Shell Nanoparticles: Synthesis, Structure Optimization and Controlled Drug Release

**DOI:** 10.3390/pharmaceutics11070357

**Published:** 2019-07-22

**Authors:** Norbert Varga, Árpád Turcsányi, Viktória Hornok, Edit Csapó

**Affiliations:** 1Interdisciplinary Excellence Centre, Department of Physical Chemistry and Materials Science, University of Szeged, Rerrich B. square 1, H-6720 Szeged, Hungary; 2Department of Physical Chemistry and Materials Science, MTA Premium Post Doctorate Research Program, University of Szeged, Rerrich B. Square 1, H-6720 Szeged, Hungary; 3MTA-SZTE Biomimetic Systems Research Group, Department of Medical Chemistry, University of Szeged, Dóm square 8, H-6720 Szeged, Hungary

**Keywords:** vitamin E, tocopherol, PLA, PLGA, core-shell nanoparticles, drug delivery, controlled drug release

## Abstract

The (±)-α-Tocopherol (TP) with vitamin E activity has been encapsulated into biocompatible poly(lactic acid) (PLA) and poly(lactide-*co*-glycolide) (PLGA) carriers, which results in the formation of well-defined nanosized (d ~200–220 nm) core-shell structured particles (NPs) with 15–19% of drug loading (DL%). The optimal ratios of the polymer carriers, the TP active drug as well as the applied Pluronic F127 (PLUR) non-ionic stabilizing surfactant, have been determined to obtain NPs with a TP core and a polymer shell with high encapsulation efficiency (EE%) (69%). The size and the structure of the prepared core-shell NPs as well as the interaction of the carriers and the PLUR with the TP molecules have been determined by transmission electron microscopy (TEM), dynamic light scattering (DLS), infrared spectroscopy (FT-IR) and turbidity studies, respectively. Moreover, the dissolution of the TP from the polymer NPs has been investigated by spectrophotometric measurements. It was clearly confirmed that increase in the EE% from ca. 70% (PLA/TP) to ca. 88% (PLGA65/TP) results in the controlled release of the hydrophobic TP molecules (7 h, PLA/TP: 34%; PLGA75/TP: 25%; PLGA65/TP: 18%). By replacing the PLA carrier to PLGA, ca. 15% more active substance can be encapsulated in the core (PLA/TP: 65%; PLGA65/TP: 80%).

## 1. Introduction

The encapsulation of the pharmaceutical ingredients in a macro- or a nanocarrier is a key factor in nanomedicine developments [1,2,3]. Utilization of drug delivery systems can increase and prolong the efficiency of the active drugs. Due to the good biocompatibility and structural properties, several materials such as proteins and mostly biodegradable polymers, e.g., chitosan, alginate, hyaluronic acid, poly(lactide) or poly(lactic acid) (PLA), polycaprolactone (PCL), poly(trimetilene-carbonate) (PTMC), etc., have been widely used as potential carriers of the nanosized drug delivery systems [4,5,6,7,8,9,10,11,12]. Many types of PLA copolymers, such as poly(lactide-*co*-glycolide) (PLGA), polylactide-poly(ethylene glycol) (PLA-PEG), poly(caprolactone-ethylene glycol-lactide) (PCELA), etc., are also well-known as drug carriers [13,14,15,16,17]. As a result of copolymerization, the hydrophilicity of the PLA can be remarkably tuned, which opens the possibility of encapsulating active ingredients that have hydrophilic or hydrophobic character into the polymer matrix. 

The (±)-α-Tocopherol (TP) is one of the highest biological activity fat-soluble vitamins from the vitamin E family and is composed of eight tocopherols and tocotrienols (alpha (α), beta (β), gamma (γ) and delta (δ) isomers for both cases) [18]. Several studies focus on the encapsulation of the TP, because vitamin E can prevent and treat many chronic and age-related diseases, e.g., Alzheimer’s disease [19]. Furthermore, the TP is described as functioning as an antioxidant, and the higher vitamin E content is associated with a lower risk of several cancer diseases (kidney, lung, bladder, etc.). 

Alqahtani et al. synthesized TP-loaded PLGA50 NPs with an average size of ca. 130 nm using Polyvinyl alcohol (PVA) as a stabilizer, but only 4–4.5% of drug loading (in mg drug/100 mg PLGA) was achieved [20]. However, Zigoneanu and coworkers increased the TP loading to 8–16% using the PLGA50 carrier and PVA and sodium dodecyl sulphate (SDS) surfactant stabilizing agents, but it was confirmed that the 86% of the drug dissolved for NPs (d ~220–280 nm) with 8% TP loading, while 36% of the TP drug released from the NPs when the loading was 16% after 1 h [19]. However, Astete et al. synthesized Span80-stabilized TP-containing PLGA50-based nanosized particles in the range of 150–200 nm and the effect of salt concentration on the size and morphology of the NPs has been interpreted, but data for the drug loading and release were not presented [21]. Murugeshu and coworkers also synthesized chitosan/PLGA50-based TP-containing NPs with 8%, 16% and 24% initial loading, but the EE% was only 45–50% and the release studies were carried out in gastrointestinal conditions (pH ~1.50) [22]. Simon et al. successfully encapsulated the TP into PLGA50 using PVA, but only 2.5 mg drug/100 mg NPs can be obtained [23]. 

In our previous work, the TP as well as the water-soluble derivative of TP (D, α-Tocopherol polyethylene glycol 1000 succinate (TPGS)) and the non-steroidal anti-inflammatory ketoprofen (KP) have been successfully encapsulated into PLA and PLGA75 (lactide:glycolide ratio 75:25) and PLGA65 (lactide:glycolide ratio 65:35) nanocarriers [13]. Instead of the commonly used and above mentioned PLGA50, the PLGA65 and PLGA75 derivatives have been firstly used as carriers for the encapsulation of TP, and the hydrophilicity properties of the carriers and the model drugs on the EE% have been studied in detail. The release measurements of nanocomposites including hydrophobic drugs are very difficult to carry out. In order to facilitate the above-mentioned studies, stabilizing agents such as surfactants have been widely used [5,19]. Non-ionic poloxamer Pluronic F127 was applied previously, which stabilized the drug-containing polymeric NPs by increasing the biocompatibility. The optimal ratios of the carrier, the drug and the stabilizing agent have a dominant effect on the structure and the EE%, as well as the controlled drug release process of the nanosized drug delivery systems, which were not studied previously. 

In the present work, the results of the TP-containing nanocomposites, using PLA, PLGA65 and PLGA75 carriers, have been completed and the determinative role of the concentration of the polymer, the TP and the PLUR stabilizing agent on the structure of the drug-containing nanocomposites has been investigated. Moreover, the drug release studies of the prepared TP-containing PLA, PLGA65 and PLGA75 core-shell NPs have also been interpreted. 

## 2. Materials and Methods

### 2.1. Materials

Polylactide (PLA, Mw = 72,200 ± 15,000 Da) and two poly(lactide-*co*-glycolide) (PLGA) derivatives with a lactide to glycolide ratio at 75:25 (PLGA75, Mw = 69,900 ± 4000 Da) and at 65:35 (PLGA65, Mw = 93,000 ± 1000 Da) were synthetized according to the previously published procedure [13]. Pluronic F127 (PLUR), (±)-α-tocopherol (TP) and sodium phosphate monobasic monohydrate (NaH_2_PO_4_ · H_2_O, ≥99%) were obtained from Sigma Aldrich (Budapest, Hungary). Sodium phosphate dibasic anhydrous (Na_2_HPO_4_, ≥99%) and sodium chloride (NaCl, ≥99%) were purchased from Molar Chemicals (Halásztelek, Hungary). All other reagents and solvents were of analytical grade and used without further purification. The deionized water was obtained by Millipore purification apparatus (18.2 MΩ cm at 25 °C). 

### 2.2. Preparation of TP-Loaded PLA/PLGA NPs

The TP-loaded PLA/PLGA NPs were prepared by nanoprecipitation method (Figure 1). The detailed experimental conditions were presented previously [13]. Briefly, the PLA/PLGA and TP with increasing concentrations (see Table 1) were dissolved in 1.5 mL of acetone, which was dropped slowly (10 μL/5 s) into the aqueous solution of the PLUR stabilizer (15 mL) under room temperature using magnetic stirring with 1000 rpm. In the interest of the evaporation of acetone, the prepared samples were further stirred (350 rpm) for two days. The dispersion was centrifugated with 40 mL MQ water at 12,000 rpm (*t* = 15 min, *T* = 25 °C). After the supernatant was removed, the NPs were redispersed in MQ water and the washing methods were repeated two times. The obtained NPs samples were freeze-dried by liquid nitrogen and lyophilized (by Christ Alpha 1-2 LDplus apparatus). 

### 2.3. Characterization Methods

The particle size was determined by dynamic light scattering (DLS) with Horiba Sz-100 (HORIBA Jobin Yvon, Longjumeau, France) equipped with a diode pumped frequency doubled (532 nm, 10 mW) laser. The measurements were carried out at 25 ± 0.1 °C with 90° of detection angle in every case. The transmission electron microscopy (TEM) images were obtained by Jeol JEM-1400plus equipment (JEOL Ltd., Tokyo, Japan) at 120 keV accelerating voltage. The Fourier transform infrared (FT-IR) spectra of the PLA/PLGA-TP NPs were registered by a Jasco FT/IR-4700 with ATR PRO ONE Single-reflection accessory (ABL&JASCO, Budapest, Hungary). The experiments were performed at room temperature from 600 cm^−1^ to 3600 cm^−1^. The resolution of the spectra was 2 cm^−1^, which was determined by 128 interferograms. 

The particles were dissolved in 1,4-dioxane to determine the EE% and DL% of the composites. The absorbance spectra of the prepared solutions were measured by a Shimadzu UV-1800 UV-Vis double beam spectrophotometer. The spectra were registered in the range of 200–500 nm using 1 cm quartz cuvette at room temperature. The characteristic absorbance band of the TP appeared at 294 nm. The concentration of the encapsulated drug was determined from the calibration curve (Appendix A). The data of the EE% in *w*/*w*% were calculated by the total drug mass used in the synthesis of the NPs (Equation (1)), while DL% was calculated by the mass of the NPs (Equation (2)).
(1)$$EE\%=\frac{encapsulated\;mass\;of\;drug}{total\;mass\;of\;drug\;in\;synthesis}\times 100$$
(2)$$DL\%=\frac{encapsulated\;mass\;of\;drug}{total\;mass\;of\;the\;nanoparticles}\times 100$$

### 2.4. Determination of the Solubility Properties of TP Drug

In order to determine the solubility of TP, 100 μL of acetone solutions of TP (5 mg·mL^−1^) was added dropwise to 10 mL of PLUR solutions (0.1–1.2 mg·mL^−1^), and the turbidity was followed with a Precision Bench Turbidity Meter LP2000 (Hanna Ins. Service Kft., Szeged, Hungary). The experiments were performed in a pure aqueous medium and in a phosphate (PBS) buffer (pH = 7.4, 0.9% NaCl) solution at 25 °C and 37 °C. 

### 2.5. Critical Micelle Concentration (cmc) Studies

The critical micelle concentration (cmc) of the applied PLUR was determined by inverse titration method in Krüss K100MK2 type surface tension equipment. The computer-controlled apparatus was supplied with a thermostat and an automatic burette. The Wilhelmy-plate method was applied. A volume of 50 mL of a 1.6 mM surfactant solution was titrated with MQ water or PBS in an aliquot of 10 mL in 40 steps. Each experimental point was the average of at least 5 measurements.

### 2.6. In Vitro Release Study

The in vitro studies of the different TP-loaded PLA/PLGA NPs were carried out by a UV-Vis spectrophotometer (500–200 nm). The release experiments were performed at 37 °C and a PBS buffer (pH = 7.4, NaCl 0.9%) containing 1 mg·mL^−1^ of PLUR was used, which facilitated the easier feasibility of the release studies [5]. The TP-loaded samples were placed into the cellulose membrane (Sigma Aldrich) with 5 mL of PLUR/PBS medium inserted into 35 mL of a dissolution phase. During the measurements, 3 mL of the release media were taken at specified intervals to measure the released concentration of TP at 268 nm. 

The release curves of the TP can be fitted by different kinetic models, such as the first order, Korsmeyer–Peppas, Peppas–Sahlin and Weibull models [24,25,26,27]. The measured points were fitted with a nonlinear regression by the QtiPlot 0.9.8.9 svn 2288 program. During the calculation session, the program finds the best fitting function for the measured points. The results consist of the fitted parameters, their standard deviation and the goodness of fitting (root mean squared error).

Depending of the release chemical condition (such as temperature, buffer solution, ionic strength, etc.), the shape of the polymers and the solubility of the drugs, the dissolution curves can be described with different kinetic models. For our calculation, the following kinetic models were used. 

First order equation is a frequently used kinetic model. This formula is well applicable for drugs where the dissolution is continuously changing over time and depends only on the concentration.
(3)$$c_{t}=ce^{-kt}$$
where *c_t_* is the concentration of the solid drug in the matrix of the carried system at *t* time, *c*_0_ is the initial concentration of the drug and *k* is the first order release constant. 

The Korsmeyer–Peppas kinetic formula is a semi-empirical power law equation where the shape of the polymer matrix (such as film, cylinder or sphere) can be taken into account in the release curve.
(4)$$\frac{c_{t}}{c_{0}}=k_{m}t^{n}$$
where *c_t_* is the concentration of the dissolved drug at *t* time, *c*_0_ is the initial concentration of drug, *k_m_* is the kinetic constant and n is the diffusion dissolution index (for sphere shaped particles *n* = 0.42 for the diffusion-controlled mechanism, *n* = 1 for the Case II relaxation controlled mechanism and 0.42 ≤ *n* ≤ 1 for both of them).

The Weibull equation is a general empirical formula which can be used for all release profiles.
(5)$$c_{t}=1-exp\left(\frac{-{\left(t-T_{i}\right)}^{b}}{a}\right)$$
where *c_t_* is the concentration of the released component in *t* time, *T_i_* is the lag time between the initial of measurement and the release of drug (in most cases *T_i_* = 0), *a* is the time scale of the process and *b* is the shape parameter (shape of the release curve is exponential if *b* = 1, parabola if *b* < 1 or sigmoid if *b* > 1).

The kinetic formula reported by Peppas and Sahlin specifies the diffusion and the relaxation contribution in the drug dissolution process.
(6)$$\frac{c_{t}}{c_{0}}=k_{1}t^{m}+k_{2}t^{2m}$$
where *c_t_* is the concentration of the dissolved drug in the *t* time, *c*_0_ is the initial concentration of the drug and *k*_1_, *k*_2_ and *m* are constants: *k*_1_ is the Fick diffusion contribution, *k*_2_ is the Case II relaxation contribution and *m* is the diffusion exponent (sphere shaped: *m* = 0.43, Fick diffusion mechanism; *m* = 0.85, Case II relaxation transport mechanism; 0.43 ≤ *m* ≤ 0.85, anomalous transport mechanism).

## 3. Results

### 3.1. Effect of the Component Concentrations on the Core-Shell Structure 

In order to determine the role of the component quantities of the nanocomposites on the size and the structure, as well as on the EE% the concentration, only one building block (polymer carrier, drug or stabilizing agent) has been modified during the synthesis while the other parameters have been kept constant. The average particle diameters of the NPs were measured by DLS, and the results are summarized in Table 1. In case of PLA concentration dependence, regardless of the amount of the TP (c = 2.5 mg·mL^−1^), the diameter of the NPs permanently increases from 120 nm (c_PLA_ = 1.25 mg·mL^−1^ in aceton phase) to ca. 200 nm (c_PLA_ = 10 mg·mL^−1^ in the acetone phase). The morphology and the structure of the NPs have been investigated by TEM images as well (Figure 2). The images clearly represent that the structure of the TP/PLUR NPs in the absence of a polymer shows less amorphous structures and TP crystal-like objects are observed. Moreover, we established that an increase in the polymer concentration results in the formation of a well-defined core-shell structure at c_PLA_ = 10.0 mg·mL^−1^. At lower polymer concentrations, this structure is not formed, and because of the low polymer concentration, the purification (centrifugation) of the NPs is impracticable; thus, the determination of the EE% was not possible. For TP that is concentration-dependent, the diameters show a slightly increasing tendency to 2.5 mg·mL^−1^, but for a 5 mg·mL^−1^ amount of TP, a higher size is obtained (d_DLS_ = 252 nm) (Table 1). The TEM images also confirm this observation (Figure 2). The well-defined core-shell structure is formed at 2.5 mg·mL^−1^ of TP quantity. With a further increase in the TP amount, the core, including the drug, shows a rather crystallized structure instead of the previously confirmed amorphous form. The EE% and the drug loading were determined for all composites. We obtained that the value of the EE% decreased from 91.28% (c_TP_ = 0.5 mg·mL^−1^) to 66.15% (c_TP_ = 5 mg·mL^−1^), while the DL% increased from 4.36% (c_TP_ = 0.5 mg·mL^−1^) to 24.85% (c_TP_ = 5 mg·mL^−1^). Considering the expected size of the NPs for optimal nanosized drug delivery systems (ca. 200 nm) as well as the crystallization of the core, the 2.5 mg·mL^−1^ amount of TP will be used for further studies at 10.0 mg·mL^−1^ of PLA polymer concentration. Besides optimizing the amount of carrier and active drug, the ratio of the stabilizing PLUR surfactant was also studied. It was clearly confirmed that in the absence of PLUR, the TP molecules were not capsulated into the polymer core, only the binding of the TP drugs onto the surface of the polymer shell is observed (Figure 2). Furthermore, it was established that the smallest particle diameter (d = 178 nm), as well as the highest DL% (ca. 20%), is obtained at 0.05 mg·mL^−1^ of PLUR concentration. Increase in the PLUR amount resulted in a higher particle size (212 nm) as well as a lower DL% values (9%). Most probably, the increase in the PLUR concentration facilitated the solubility of the hydrophobic TP, thus hindering the encapsulation process. In order to confirm the above-mentioned phenomena, turbidity measurements were carried out (Appendix A). According to the TEM images (Figure 2) and the turbidity studies (Appendix A), we can conclude that a low quantity (0.1 mg·mL^−1^) of PLUR surfactant is advantageous for the formation of PLA-based core-shell NPs, but the presence of a higher amount of PLUR (>0.1 mg·mL^−1^) results in the decrease of the EE%. Moreover, the presence of a higher amount of stabilizer (≥1.0 mg·mL^−1^) causes aggregation (Figure 2). 

Besides the determination of the optimal ratio of the composite building blocks, the TP was encapsulated in PLGA75 and PLGA65 copolymers that have increasing hydrophilicity. During the NPs synthesis, the previously optimized concentrations of the polymer carrier (10.0 mg·mL^−1^), the PLUR (0.1 mg·mL^−1^) and the TP (2.5 mg·mL^−1^) were used. Based on DLS studies, we established that the particle size increases from 203 nm to 226 nm with a decrease of the lactide part in the polymer (Figure 3A). Using the optimized component quantities, the core-shell structure was confirmed for PLGA75 and PLGA65-based TP-containing NPs (Figure 3B). The EE% and DL% has been determined for the PLGA75 and PLGA65-based system as well, and we observed that the replacement of the PLA carrier to PLGA75 and PLGA65 polymers resulted in the increase in the EE% to 75.72% and 87.69%, respectively. In addition, the drug loading also increased from 14.73% (PLA) to 15.92% (PLGA75) and to 17.98% (PLGA65) with decreasing lactide content. The higher EE% and DL% can be explained by the fact that the precipitation of the more hydrophilic PLGA carriers is slower than that for PLA and TP, which helps the formation of the well-defined core-shell structure [13].

### 3.2. Structural Characterization of the TP-Loaded PLA and PLGA Core-Shell NPs

To determine the interaction between the PLA and TP, the composites have been examined by infrared spectroscopy measurements. Figure 4 displays the spectra of the PLA-based NPs in the absence (Figure 4A) and in the presence of TP at different concentrations (Figure 4B,C). The TP-sensitive bands appear in the range of 3050–2800 cm^−1^ and 1150–1000 cm^−1^. It is obvious that the intensity of all the determinative bands systematically increase by increasing TP content. At 2994, 2944 and 2968 cm^−1^, the asymmetric and symmetric CH stretching vibrations of the –CH_2_ and –CH_3_ groups of the drug appear. Due to the increasing TP concentration, these bands become more intense, indicating the presence of TP molecules in the polymer NPs. The carbonyl group of the PLA appears at 1750 cm^−1^ [28,29]. The TP does not contain a C=O group, thus this band could originate only from the polymer. In the fingerprint region (ν ≤ 1500 cm^−1^), the deformation and bending vibration of the –CH_3_ and –CH_2_ groups appear at 1453, 1381 and 1267 cm^−1^, while the band at 1181 cm^−1^ attributes to the stretching mode of the C–O–C (ester). At around 1086 cm^−1^, further bands of the C–O–C stretching vibration can be observed, which have shoulders (symmetric and asymmetric stretching vibrations). Because of the Ar–O–C group in the TP, the intensity of this C–O–C symmetric stretching vibration at around 1050 cm^−1^ is increased. The band at 865 cm^−1^ and 751 cm^−1^ is characteristic of the polymer carrier, and no shift is observed. The IR measurements performed for PLGA75 and PLGA65 resulted in similar spectra. A strong irreversible interaction between the PLA (or PLGA75, PLGA65) and the TP cannot be discovered (Appendix A), which facilitates the spontaneous release of the TP active drug from the polymer NPs, but the presence of the TP in the different composites were definitely confirmed by IR. 

Turbidimetric measurements were performed to investigate the interaction between the TP and the PLUR stabilizing agent in a pure aqueous solution and in PBS at 25 °C and 37 °C (pH = 7.4, 0.9% NaCl); the results are presented in Figure 5. In MQ water (Figure 5A), the turbidity of TP is systematically decreased till ca. 0.9 mg·mL^−1^ of PLUR concentration. Over time, more and more TP can dissolve in this medium. In contrast, in the PBS solution at 25 °C and at 37 °C, the titration curves exhibit steeper decreasing intensity, which may be due to the reduced critical micellization concentration (cmc) of PLUR in the presence of salt. In the case of the PBS solution, the turbidity remains constant from 0.7 mg·mL^−1^ at 25 °C, while an increase in the temperature to 37 °C 0.6 mg·mL^−1^ value is observed. It is important to mention that the solubility of the TP scarcely depends on time at 37 °C, which allows the possible use of the PLUR stabilizing agent for in vitro drug release measurements. 

It is well known that the surfactant affects the solubility of the TP drug in the absence of a polymer carrier. Namely, above cmc, due to the micellization ability of the PLUR, the solubilization of the drug dominates forming TP-loaded individual micelles, while below cmc, only the solubility increases. Accordingly, the cmc of the PLUR was measured by surface tension measurements (Figure 6). In the literature, very different values can be found; the obtained range of cmc is 2–7 mg·mL^−1^ [30,31]. Similar values were measured, but we determined that at 25 °C, the obtained cmc is decreased from 4.92 mg·mL^−1^ to 1.99 mg·mL^−1^ in the presence of the phosphate buffer. If the temperature rises from 25 °C to 37 °C, these values are further decreased. We can conclude that the concentration of the PLUR in the composites is significantly lower than the cmc (4.92 mg·mL^−1^) value, which excludes the presence of TP-loaded individual micelles. 

### 3.3. In Vitro Drug Release Experiments 

The determination of the exact TP amount released from the different composites was carried out by the UV-Vis spectrophotometric method. The spectra of the TP and the calibration curve are presented in Figure 7. The absorption bands of the TP appear in the UV range. 

After the characterization of the TP-loaded PLA and PLGA NPs, the mechanism of the drug release was investigated. The drug dissolution profiles and the fitting of these curves by different kinetic models are demonstrated in Figure 8 (Appendix A). The suitability of several models like Korsmeyer–Peppas, Peppas–Sahlin, the first-order and the Weibull models were investigated. The release curves clearly show that a high amount of TP is retained in the polymers after 7 h. It was also observed that in the first half an hour, the dissolution of the drug occurs relatively quickly, but after that, measurable slow dissolution is observed. Moreover, we found that the active substance is released slowly with a decrease of the lactide part (PLA, 35.0%, PLGA75, 28.3%, PLGA65, 19.8%) in 7 h. The slowest dissolution occurred in the carrier-free TP (15%). Because of the higher hydrophobicity, a higher amount of the non-encapsulated drug can be attached to the surface of the particles, which confirms the above-mentioned dissolution order [13]. Thanks to the application of the nanosized drug carrier systems (polymer NPs), the TP molecules can bind to the enhanced specific surface area of the NPs, which facilitates the dissolution of more TP, in contrast to the bulk TP.

Taking into account the coefficient of the determination (*R*^2^), the Peppas–Sahlin model was the best kinetic formula for our systems (Table 2). The values of the Case II relaxation contribution (*k*_2_) were negative in all cases; therefore, this model does not provide complete information about the dissolution of the TP, but the low diffusion exponent (*m*) is referred to for the Fickian release. 

The second-best kinetic model was the Korsmeyer–Peppas model where the diffusion dissolution index (*n*) gives the information about the diffusion and the erosion of the matrix. The values of the *n* are increased from *n* = 0.182 (PLA) to *n* = 0.223 (PLGA65), which is referred to for the diffusion-controlled quasi Fickian drug release. Furthermore, it is important to note that the value of the diffusion dissolution index is lower than 0.42. This is caused by the high polydispersity of the particles (because they are lyophilized) and the very low degradation of the polymers [32]. Presumably, the Case II relaxation contribution from the Peppas–Sahlin model will be low by these effects. 

During the slow degradation of the PLA/PLGA carrier, the drug diffuses with difficulty from the core of the particles. The measured and calculated results clearly stated that the released drug originates from the surface region of the particles. Based on this, we could calculate the approximate quantity of the TP in the core of the particles after 7 h: 65.0% (PLA), 71.7% (PLGA75) and 80.2% (PLGA65) from the encapsulated mass of the TP; therefore, a significant amount of the active ingredient can be encapsulated inside the particles.

## 4. Conclusions

In the work presented here, the successful encapsulation of the hydrophobic α-Tocopherol, one of the determinative natural forms of vitamin E, was carried out using the PLA, PLGA75 and PLGA65 biocompatible polymer carriers by increasing hydrophilicity. To the best of our knowledge, we first proved the formation of well-defined nanosized TP-core PLA/PLGA-shell structured nanocomposites. Optimization of the experimental conditions, such as optimal concentration of drug, polymer carrier and stabilizing PLUR non-ionic surfactant, resulted in the formation of core-shell NPs within the diameter range of ~200–220 nm. For the PLA-based system 14.7% of drug loading was achieved, where most of the TP molecules are encapsulated in the core (65%), while the remaining part of the active ingredient is located on the surface of the polymer shell. By replacing the hydrophobic PLA to PLGA copolymers, both the drug loading (PLGA75: 16%, PLGA65: 18%) as well the EE% can be increased (PLA/TP: 69.1%; PLGA75/TP: 75.7%; PLGA65/TP: 87.7%). Furthermore, the PLGA-based composites contain 71.7% and 80.2% of the encapsulated TP in the core. Considering the slower precipitation ability of the PLGA copolymers in contrast to the PLA and the active ingredient, the higher encapsulation efficiency can be explored. Besides the preparation and the characterization of the composites, the drug release was also studied. The dissolution curves clearly show that depending on the polymer, more than 65–80% of the TP is contained in the composites after 7 h. In the first half an hour, the dissolution of the drug occurs relatively quickly, but after that, measurable slow dissolution was observed. Moreover, we found that the active substance is released slowly with decreasing lactide part (PLA, 35.0%; PLGA75, 28.3%; PLGA65, 19.8%) in 7 h. Thanks to the biocompatibility, cost-effectivity and tuneable hydrophilic properties, the PLA/PLGA polymers are potential candidates for controlled drug release and for the encapsulation of hydrophobic TP or similar sized and structured molecules in core-shell nanosized particles.

## Figures and Tables

**Figure 1 pharmaceutics-11-00357-f001:**
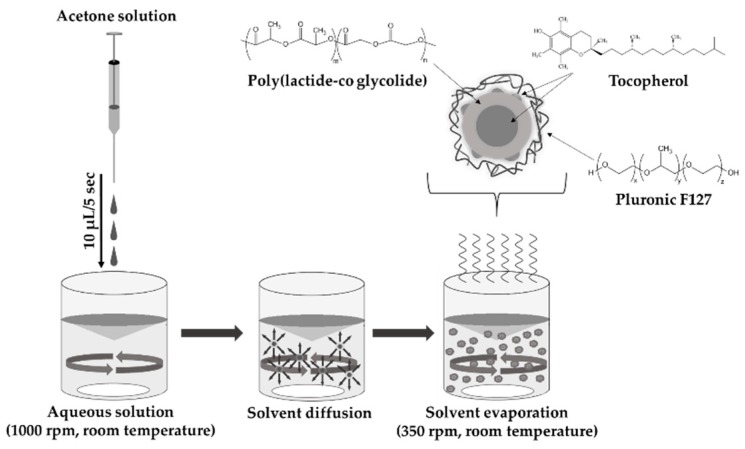
Schematic representation of the preparation of TP-loaded PLGA NPs stabilized by PLUR using nanoprecipitation technique.

**Figure 2 pharmaceutics-11-00357-f002:**
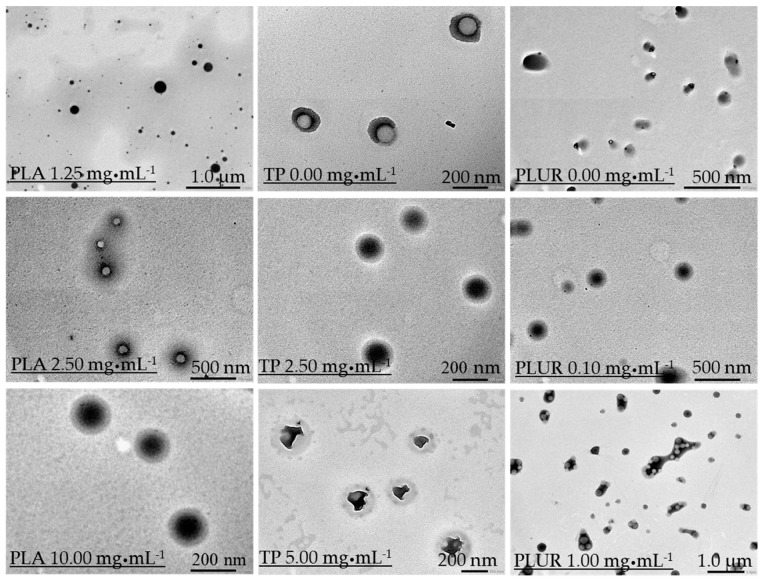
Representative TEM images of the TP-containing PLA NPs using different component concentrations (PLA: 1.25–10.0 mg·mL^−1^; TP: 0–5.0 mg·mL^−1^ and PLUR: 0–1.0 mg·mL^−1^).

**Figure 3 pharmaceutics-11-00357-f003:**
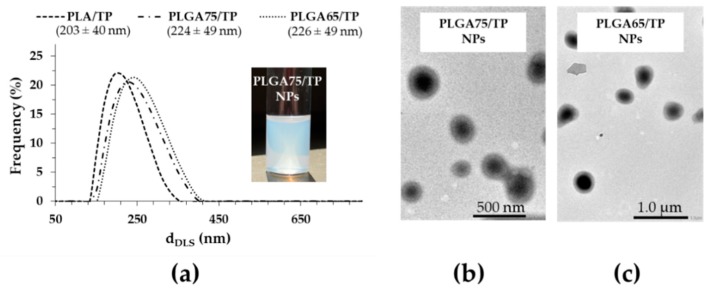
The particle size distribution of the TP-loaded PLA and PLGA NPs (**a**), and representative TEM images of the PLGA75–(**b**) and PLGA65–(**c**) based composites.

**Figure 4 pharmaceutics-11-00357-f004:**
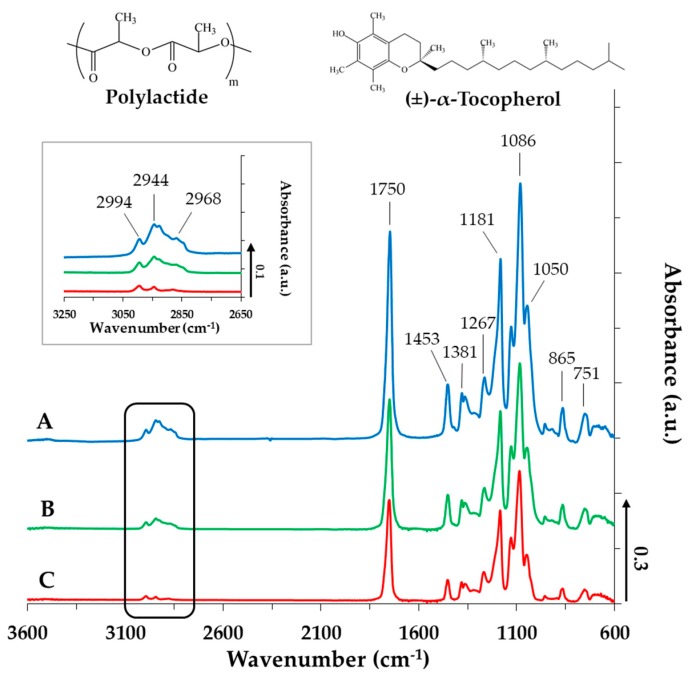
IR spectra of the TP-loaded PLA NPs (A: c_TP_ = 0 mg·mL^−1^; B: c_TP_ = 2.5 mg·mL^−1^; C: c_TP_ = 5 mg·mL^−1^, c_PLA_ = 10 mg·mL^−1^ and c_PLUR_ = 0.1 mg·mL^−1^).

**Figure 5 pharmaceutics-11-00357-f005:**
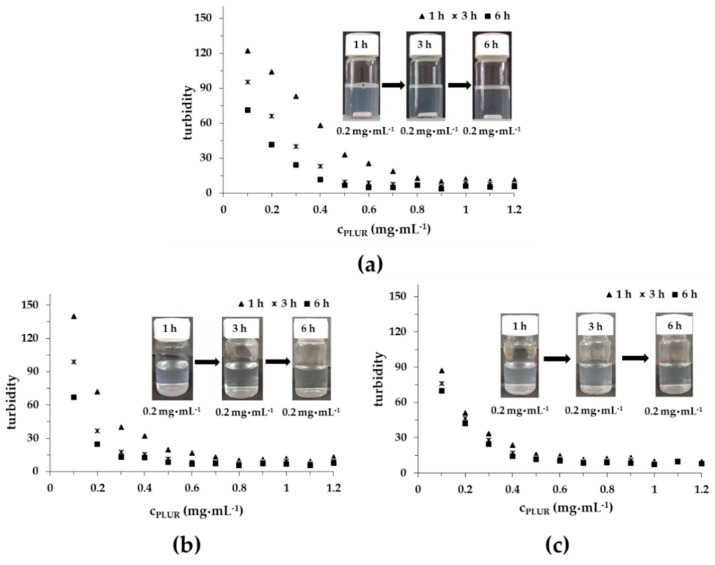
The turbidity of the TP in the PLUR solution at 25 °C in an aqueous medium (**a**), in a PBS buffer at 25 °C (**b**) and at 37 °C (**c**) (pH = 7.4, 0.9 *w*/*w*% NaCl) (c_TP_ = 0.05 mg·mL^−1^).

**Figure 6 pharmaceutics-11-00357-f006:**
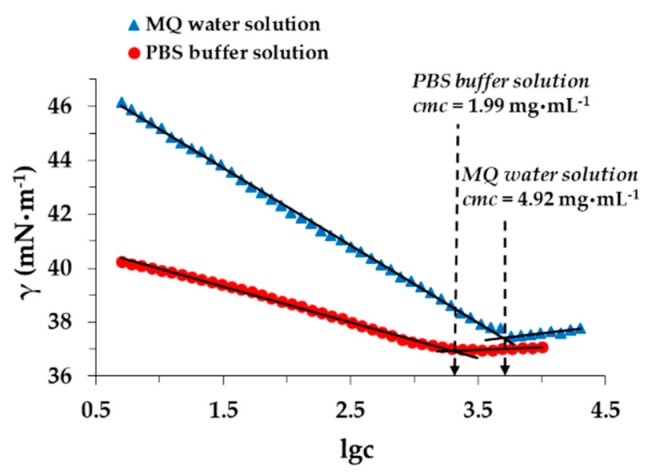
Determination of PLUR cmc at 25 °C in MQ water and in a PBS solution.

**Figure 7 pharmaceutics-11-00357-f007:**
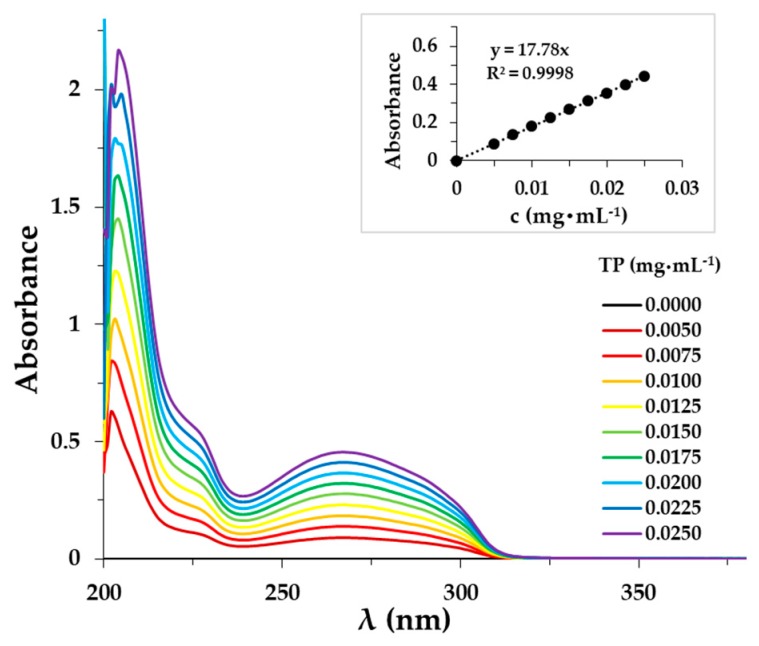
UV spectra of the TP in the PBS solution (c_PLUR_ = 1.0 mg·mL^−1^, 0.9 *w*/*w*% NaCl).

**Figure 8 pharmaceutics-11-00357-f008:**
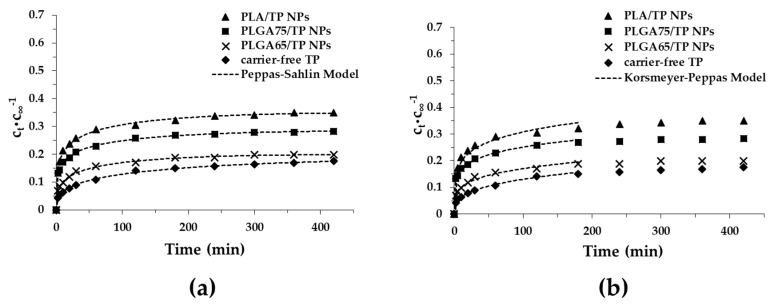
Release profiles and different kinetic models-predicted (Peppas-Sahlin model (**a**), Korsmeyer-Peppas model (**b**)) for the release curves of TP from PLA and PLGA NPs in the PBS buffer (pH = 7.4, 0.9 *w*/*w*% NaCl).

**Table 1 pharmaceutics-11-00357-t001:** The concentration of the components, the average particle diameter, the polydispersity index (PI), the encapsulation efficiency (EE%) and the drug loading (DL%) of the TP-loaded PLA NPs.

	Acetone Phase		Aqueous Phase				
Sample	c_PLA_ (mg·mL^−1^)	c_TP_ (mg·mL^−1^)	c_PLUR_ (mg·mL^−1^)	d_DLS_ ± SD ^1^ (nm)	PI ± SD	EE%	DL%
**PLA concentration dependence**	1.25	2.5	0.1	120 ± 33	0.120 ± 0.043	–	–
	2.5	2.5	0.1	156 + 28	0.039 ± 0.012	–	–
	5.0	2.5	0.1	179 ± 35	0.082 ± 0.049	–	–
	10.0	2.5	0.1	201 ± 38	0.095 ± 0.026	69.11	14.73
**TP concentration dependence**	10.0	0	0.1	188 ± 37	0.092 ± 0.059	–	–
	10.0	0.5	0.1	189 ± 34	0.048 ± 0.028	91.28	4.36
	10.0	1.0	0.1	192 ± 30	0.062 ± 0.032	75.61	7.02
	10.0	2.5	0.1	201 ± 38	0.095 ± 0.026	69.11	14.73
	10.0	5.0	0.1	252 ± 53	0.073 ± 0.033	66.15	24.85
**PLUR concentration dependence**	10.0	2.5	0	179 ± 40	0.315 ± 0.040	72.19	15.29
	10.0	2.5	0.05	178 ± 21	0.304 ± 0.095	98.34	19.73
	10.0	2.5	0.1	201 ± 38	0.095 ± 0.026	69.11	14.73
	10.0	2.5	0.5	206 ± 36	0.089 ± 0.065	57.94	12.65
	10.0	2.5	1.0	212 ± 36	0.066 ± 0.029	40.75	9.24

^1^ The experimental error of the peak maximum is below 2.5%.

**Table 2 pharmaceutics-11-00357-t002:** Determined parameters of the TP release by fitting several kinetic equations.

**Peppas–Sahlin Formulation**	***k*_1_ (min^–*m*^)**	***k*_2_ (min^–2*m*^)**	***m***	***R*^2^**
PLA/TP NPs	0.1371	−0.01346	0.260	0.9974
PLGA75/TP NPs	0.1165	−0.01189	0.248	0.9986
PLGA65/TP NPs	0.0572	−0.00412	0.317	0.9974
TP	0.0304	−0.00124	0.366	0.9978
**Korsmeyer–Peppas Formulation**	***k_m_* (min^–*n*^)**	***n***	***R*^2^**	
PLA/TP NPs	0.1339	0.182	0.9898	
PLGA75/TP NPs	0.1120	0.175	0.9982	
PLGA65/TP NPs	0.0592	0.223	0.9904	
TP	0.0310	0.311	0.9977	
**Weibull Formulation**	***a***	***b***	***R*^2^**	
PLA/TP NPs	6.52	0.178	0.9896	
PLGA75/TP NPs	7.85	0.166	0.9819	
PLGA65/TP NPs	15.45	0.220	0.9898	
TP	28.19	0.285	0.9945	
**First Order Formulation**	***k* (min^–1^)**	***R*^2^**		
PLA/TP NPs	0.0094	0.9792		
PLGA75/TP NPs	0.0113	0.9532		
PLGA65/TP NPs	0.0130	0.9514		
TP	0.0079	0.9856		

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
