# Peer review of "Vitamin E-Loaded PLA- and PLGA-Based Core-Shell Nanoparticles: Synthesis, Structure Optimization and Controlled Drug Release"

_pharmaceutics, 2019, doi:10.3390/pharmaceutics11070357_

Round 1

Reviewer 1 Report

Abstract:

Line 20: define the range of obtained EE%  considered high.

What does Ca. means. I am personally not familiar with the term.

Line 28: does the 65% and 80% means drug loading efficiency. Please clarify in the abstract. It gets confusing while reading the abstract with EE% and DL%.

Main text:

Line 69-70: ‘The release measurements of nanocomposites including hydrophobic drugs are very difficult to carry out. In order to facilitate the above-mentioned studies stabilizing  agents such as surfactants have been widely used.; Does the above-mentioned studies mean release studies? If yes, then clarify and provide a reason how surfactants help in release studies.

Line 92: replace ‘was’ with ‘were’.

Line 96-100: In the synthesis: is 15 mL of dispersion added to 40 mL of water. It not clear by text. Also, is the freeze dryer used temperature controlled or just a bench top lyophilizer with pressure control only. If latter is true, then did you observe melting of the frozen solution at such high volumes? If yes, any consequences?

Line 142: In vitro release: is the 3 mL of media removed replenished during the release study?

Line 138-140: reframe the statement. Present and past tense in the same statement.

Line 159: replace ‘low’ with ‘law

Line 192-194: the two statements are contradicting. First references a figure but next says not presented here. Please clarify.

Line 221-22: any rationale behind why higher stabilizer concentration causes aggregation?

Line 238-239: correct the grammar of the statement.

In vitro release results: please explain the release results more carefully. First spiked release is said to be released due to surface presence of TP but overall release in 7 hrs is also a surface release and not encapsulated TP release.  One needs a better rationale behind it. If this is a mis-interpretation from me, then better explanation is required. Also, what is the significance of release profile from application point of view since a major amount of release happens very slow over months.

How well was the chemical stability of TP determined in water/ PBS used for release?

Line 311: how is carrier free dissolution determined?

Figure 8: is it TP release profile curve? If yes, then what does the legend ‘TP’ represents here?

Overall I don’t understand how the released TP was determined with UV-Vis since PBS buffer was used having turbidity with the PLUR/TP combination.

Line 332-336: lots of grammatical error and confusion statement formation. Please re-frame the statements and keep it clear.

Line 357: reframe the statement.

Line 362: spell check tunable.

Overall, lot of grammatical checks are required and better explanation of the observed phenomenon.

Reviewer 2 Report

Dear authors

The present manuscript is very well written and 

Figure 1 it's ok, but it can be improved by adding experiment condition in the draw. I know is in the previous paragraph, but figure can be self-explanatory (please add stirring velocity, heating temperature, what the solution are and what the drops are)

All the characterization methodology are well chosen, but, since the application is for a drug delivery system, it is important to measure biocompatibility of the final polymeric carriers. Even if you chose a well documented FDA approved biocompatible polymers, always is important to show the that used procedure don't let any cytotoxic residue.

In the case of the obtained particle size, what is the recommended size for the TP administration, these sizes are adequate?. Taking into account the %EE and TP release, which of the evaluated polymeric nanoparticles are suitable for further studies? and what studies you propose are next for the implementation of the proposed drug delivery system. 

Please add the pertinent information answering these questions
